# A Synthetic CPP33-Conjugated *HOXA9* Active Domain Peptide Inhibits Invasion Ability of Non-Small Lung Cancer Cells

**DOI:** 10.3390/biom10111589

**Published:** 2020-11-23

**Authors:** Seong-Lan Yu, Han Koo, Se-In Lee, JaeKu Kang, Young-Hyun Han, Young Il Yeom, Dong Chul Lee

**Affiliations:** 1Priority Research Center, Myunggok Medical Research Institute, College of Medicine, Konyang University, Daejeon 35365, Korea; silove2005@gmail.com (S.-I.L.); jaeku@konyang.ac.kr (J.K.); han0531@hanmail.net (Y.-H.H.); 2Personalized Genomic Medicine Research Center, Korea Research Institute of Bioscience and Biotechnology (KRIBB), Daejeon 34141, Korea; koohan@kribb.re.kr (H.K.); yeomyi@kribb.re.kr (Y.I.Y.); 3Department of Functional Genomics, University of Science and Technology (UST), Daejeon 34113, Korea; 4Department of Pharmacology, College of Medicine, Konyang University, Daejeon 35365, Korea

**Keywords:** *HOXA9*, CPP33-HADP, *SNAI2*, cell invasion, non-small cell lung cancer

## Abstract

Homeobox A9 (*HOXA9*) expression is associated with the aggressive growth of cancer cells and poor prognosis in lung cancer. Previously, we showed that *HOXA9* can serve as a potential therapeutic target for the treatment of metastatic non-small cell lung cancer (NSCLC). In the present study, we have carried out additional studies toward the development of a peptide-based therapeutic agent. Vectors expressing partial DNA fragments of *HOXA9* were used to identify a unique domain involved in the inhibition of NSCLC cell invasion. Next, we performed in vitro invasion assays and examined the expression of EMT-related genes in transfected NSCLC cells. The C-terminal fragment (*HOXA9*-C) of *HOXA9* inhibited cell invasion and led to upregulation of *CDH1* and downregulation of *SNAI2* in A549 and NCI-H1299 cells. Reduced *SNAI2* expression was consistent with the decreased binding of transcription factor NF-kB to the *SNAI2* promoter region in *HOXA9*-C overexpressing cells. Based on the above results, we synthesized a cell-permeable peptide, CPP33-HADP (*HOXA9* active domain peptide), for lung-specific delivery and tested its therapeutic efficiency. CPP33-HADP effectively reduced the invasion ability of NSCLC cells in both in vitro and in vivo mouse models. Our results suggest that CPP33-HADP has significant potential for therapeutic applications in metastatic NSCLC.

## 1. Introduction

Lung cancer is one of the most common types of cancer worldwide and contributed to about 12% of the total number of new cases diagnosed in 2020 [1]. NSCLC accounts for 80–85% of all lung cancer diagnoses and has a 5-year survival rate of less than 18%. Clinically, therapeutic measures for treating NSCLC have been identified. There are developed targeted therapy by targeting cancer driver genes such as epidermal growth factor receptor (EGFR) and anaplastic lymphoma kinase (ALK). Immunotherapy targeting immune-modulating mechanisms are also being developed recently. Thus, advanced treatment methods have been improved the survival rate of patients with NSCLC [2].

Homeobox (*HOX*) genes encode critical proteins that regulate anterior-posterior axis formation during embryogenesis. The 39 *HOX* genes are arranged in four clusters, namely *HOXA*, *HOXB*, *HOXC*, and *HOXD*, located on four different chromosomes [3,4]. Previous studies have found that small differences in homeodomains themselves confer unique properties to *HOX* genes [5]. *HOX* genes are reported to be transcriptional activators or repressors in the progression of various cancers [6]. Of these genes, *HOXA9* is reported to be strongly associated with the aggressive growth of lung cancer cells. *HOXA9* protein is downregulated in lung cancer tissues and plays the role of a tumor progression repressor [7,8]. It is well established that *HOXA9* contains a DNA-binding homeobox domain and regulates downstream gene expression [9]. In a previous study, we also demonstrated that exogenous upregulation of *HOXA9* inhibited tumor cell invasion and migration and attenuated the expression of snail family zinc finger 2 (SNA12/SLUG) through repression of nuclear factor (NF)-kB activity and suggested that *HOXA9* can serve as a potential target for developing anticancer agents [10]. *SNAI2*/SLUG, known as the core EMT (Epithelial-to-Mesenchymal Transition) transcription factor, represses cell-cell adhesion proteins such as *CDH1*/E-cadherin, a typical epithelial maker [11]. EMT plays an essential role in the invasive and metastatic properties of cancer.

Drug delivery systems are used to transport therapeutic drugs into the cells to safely achieve the therapeutic effect. Amongst the current drug delivery systems, cell-penetrating peptides (CPPs), due to their rapid cellular uptake, have shown great potential as drug delivery agents in recent years. CPPs are usually non-toxic and are short peptides that deliver various cargo such as RNA, DNA, and chemical molecules across cellular membranes into cells. The first CPP was discovered from the trans-activating transcriptional activator (TAT) of human immunodeficiency virus 1 (HIV-1) [12,13]. CPPs are broadly classified into non-tissue specific peptides and tissue-specific peptides [14]. Polyarginine CPP uptake is reported in major tissues including the bladder, kidney, liver, and lung, using intraperitoneal or intravenous injections [15]. In tissue-specific peptides, CPP33 penetrated only NSCLC cells when tested in eight different tissues, including normal tissues [16]. Moreover, Lin et al. demonstrated that a dual-ligand liposome made with anti-carbonic anhydrase IX and CPP33 selectively penetrates NSCLC cells rather than normal lung cells [17]. Recently, several studies have focused on the potential of CPPs as carriers for the delivery of anti-tumoral proteins [18,19,20,21]. Our group also developed a CPP-conjugated recombinant protein for the effective delivery of the *HOXA9* protein into NSCLC. The recombinant cell-permeable protein reduced the motility of NSCLC cells in an in vivo experimental model [22].

However, a unique domain of the *HOXA9* protein that negatively modulates the metastatic progression of NSCLC cells is currently not known. In the present study, we identified a sequence motif associated with the inhibitory activity of *HOXA9* toward an invasion of NSCLC cells. We also investigated the therapeutic efficacy of the sequence motif using the synthetic cell-penetrating peptide CPP33-HADP (*HOXA9* active domain peptide) in in vitro and in vivo experimental models.

## 2. Materials and Methods

### 2.1. Cell Culture

Normal lung fibroblast cells (MRC-5) and non-small lung cancer cells (A549 and NCI-H1299) were obtained from the American Type Culture Collection (ATCC, Manassas, VA, USA) and Korea Cell Line Bank (KCLB) (Seoul, Korea). Cells were cultured on minimum essential media (MEM, Hyclone Laboratories, Logan, Utah, USA) for MRC-5, Dulbecco’s Modified Eagle Medium (DMEM, Hyclone Laboratories) for A549 and RPMI 1640 medium for NCI-H1299 (Hyclone Laboratories) supplemented with 10% fetal bovine serum (FBS, Hyclone Laboratories) and 1% penicillin/streptomycin (Hyclone Laboratories). The cells were maintained at 37 °C in a humidified atmosphere containing 5% CO_2_.

### 2.2. DNA Cloning, Transfection, and Peptides

To identify the sequence motif of *HOXA9* protein associated with inhibition of cell motility in NSCLC cells, the open reading frame (ORF) of *HOXA9* was subdivided into 3 sections with overlapping parts. The 3 sections of *HOXA9* were amplified using a forward primer containing an *EcoR*I restriction site and a reverse primer containing a *Not*I restriction site. *HOXA9* gene contains two exons and one intron. The *HOXA9*-N and M primer set is located at exon 1, and the *HOXA9*-C primer set corresponds to the interface region between the two exons including the intron. The primer sequences are indicated in Appendix A. These 3 amplicons (*HOXA9*-N, -M, -C) were subcloned into the vector pcDNA3 (Addgene, Watertown, MA, USA) and each clone was verified using sequencing. The NSCLC cells were transfected with the 3 fragments of *HOXA9* using Lipofectamine 2000 (Invitrogen, Waltham, MA, USA) according to the manufacturer’s recommended protocol. The peptides, C-HADP (FITC-HARSTRKKRCPSGGSTERQVKIWFQNRRMKMKKINK), R10-HADP (FITC-RRRRRRR RRR-HARSTRKKRCPSGGSTERQVKIWFQNRRMKMKKINK), and CPP33-HADP (FITC-RLWMRWYSP RTRAYGHARSTRKKRCPSGGSTERQVKIWFQNRRMKMKKINK), were synthesized through peptides synthesis service of PEPTRON (Daejeon, Korea) and fluorescein isothiocyanate (FITC) was included in each peptide to verify intracellular transduction. These peptides were >95% pure as determined by HPLC and mass spectrographic analysis. For intracellular transduction, A549 and NCI-H1299 cells were seeded on plates. After 24 h, the cells were washed twice with serum-free media and incubated with 1 µM C-HADP, R10-HADP, or CPP33-HADP in serum-free medium for the indicated time periods.

### 2.3. Cell Invasion Assay

Invasion assays were performed using 48-well microchemotaxis chambers that contained 12 µm-pore membranes (Neuroprobe, Gaithersburg, MD, USA) pre-coated with 10 µg/mL Matrigel (BD Bioscience, San Jose, CA, USA) for the invasion assay as described previously [22]. The cells were seeded in the chambers, in triplicate, together with serum-free media, and incubated for 28 h for invasion. The membranes were fixed and stained using Diff-Quik reagent (Sysmex Corporation, Kobe, Japan). The invaded and migrated cells were photographed under a light microscope, and relative invasion and migration rates were calculated based on comparison with the control.

### 2.4. Quantitative Realtime PCR (qRT-PCR) Analysis

Total RNA was isolated using TRIzol^TM^ Reagent (Invitrogen, Thermo Fisher Scientific) according to the manufacturer’s instructions. cDNA was synthesized using MMLV reverse transcriptase (Promega, Madison, WI, USA) after which qRT-PCR was performed in triplicate for *CDH1*, *SNAI2*, and *GAPDH* genes using an iQ SYBR Green Supermix and a CFX96 qPCR machine (BioRad Laboratories, Hercules, CA, USA). The used primers are as follows: forward, 5′-TCAGCGTTGTGACTGTGAA-3′ and reverse, 5′-CCTCCAAGAATCCCCAGAAT-3′ for *CDH1*; forward, 5′-TCTGCAGACCCATTCTGATG-3′ and reverse, 5′-AGCAGCCAGAT TCCTCATGT-3′ for *SNAI2* and forward, 5′-ACAGTCAGCCGCATCTTCTT-3′ and reverse, 5ʹ-ACGACCAAATCCGT TGACTC-3′ for *GAPDH*. The following amplification conditions were used: an initial denaturation step at 95 °C for 3 min, followed by 40 cycles of denaturation at 95 °C for 10 s, annealing at 56 °C for 10 s, and extension at 72 °C for 10 s. The 2^-ΔΔCt^ method was used to calculate mRNA expression levels using *GAPDH* as the reference gene.

### 2.5. NF-kB Reporter Gene Assay

A549 and NCI-H1299 cells were transfected with a luciferase reporter containing NF-κB binding sites (Clontech, Mountain View, CA, USA) as described previously [22]. Cells were harvested for luciferase assays after transfection with pcDNA, *HOXA9*-F, *HOXA9*-N, *HOXA9*-M, or *HOXA9*-C for 24 h. Luciferase assays were performed in triplicate and each experiment was repeated at least three times.

### 2.6. Chromatin Immunoprecipitation (ChIP) Assay

ChIP assay were performed with the ChIP assay kit (Millipore, Burlington, MA, USA) according to the manufacturer’s protocol. The NF-kB binding site in the *SNAI2* promoter region was detected by means of PCR performed using primer sets as described previously [23].

### 2.7. Cell Cytotoxicity Assay

To determine cell cytotoxicity of the cell-penetrating peptide, A549 and NCI-H1299 (5 × 10^3^) cells were seeded in 96-well tissue culture plates, and then the cells were treated with 1 µM CPPs for 48 h. Cell viability was measured with MTT reagent ((3,4,5-dimethylthiazol-2yl)-5-diphenyl-tetrazolium bromide, Sigma-Aldrich, St. Louis, MO, USA). Briefly, the surviving cells were treated with 500 μg/mL of MTT solution for 2 h, after which the absorbance was measured at 540 nm. The survival rate was calculated as the ratio of the absorbance of the treated wells to that of the control wells.

### 2.8. Intracellular Transduction of Synthetic Cell-Penetrating Peptide

Cells grown on coverslips were cultured overnight and were then treated with 1 µM C-HADP, R10-HADP, or CPP33-HADP for 4 h. Cells were fixed with 4% formaldehyde, washed with phosphate-buffered saline (PBS), and permeabilized with 0.3% triton-X100 (Sigma Aldrich). Next, the cells were stained with 4′,6′-diamidino-2-phenylindole dihydrochloride (DAPI; Invitrogen). The stained cells were examined using a confocal microscope (LSM710; Carl Zeiss, Oberkochen, Germany). To identify transduction of peptides in the cytoplasm, cells were cultured on a plate overnight and the following day 1 µM C-HADP, R10-HADP, or CPP33-HADP were added and incubated for 4 h. Cells were trypsinized and fixed with 4% formaldehyde. Cells were treated with 0.4% trypan blue to remove fluorescence bound with the cell membrane and then washed with PBS. The fluorescence intensity was determined with a cytoFLEX cytometer (Beckman Coulter Life Science, Indianapolis, IN, USA).

### 2.9. In Vivo Metastasis

To determine the role of the peptides in vivo, female BALB/c-nu/nu mice were purchased from the Central Animal Laboratory (SLC, Shizuoka, Japan). Distant metastatic tumors were developed through tail vein injection of A549-Luc cells (4 × 10^6^), and 4 mg/kg synthetic C-HADP or CPP33-HADP peptide was intraperitoneally injected twice a week for 4 weeks. The extent of lung cancer metastasis was periodically imaged by bioluminescence imaging (CLS136331, PerkinElmer, MA, USA). Animal experiments were approved by the Animal Research Ethics Committee of the Korea Research Institute of Bioscience and Biotechnology (KRIBB).

### 2.10. Hematoxylin and Eosin Staining and Immunohistochemical Analysis

Animals were sacrificed following institutional guidelines, after which tumors from lung tissues were removed and fixed overnight with 10% formalin at room temperature. The fixed tissues were paraffin-embedded, after which 4 μm sections were placed on silylated slides (Histoserv, Inc., Germantown, MD, USA) and then stained with hematoxylin, followed by eosin. To determine SLUG expression in lung tissues, we performed immunohistochemical analysis using a primary antibody for SLUG (Abcam, Cambridge, UK) and secondary Alexa 594 conjugated antibodies (Invitrogen, Thermo Fisher Scientific) as described previously [22]. The stained cells were visualized using a confocal microscope (LSM710; Carl Zeiss, Oberkochen, Germany).

### 2.11. Statistical Analyses

All graphical data are presented as means ± standard deviation (SD). The results were analyzed using Student’s *t*-test; *p* < 0.05 or *p* < 0.01 was considered statistically significant.

## 3. Results

### 3.1. HOXA9-C Fragment Inhibits Cell Invasion in NSCLC Cells

Our research group has previously reported that *HOXA9* overexpression inhibits tumor aggressiveness in NSCLC cells [10,22]. However, the sequence motif of *HOXA9* associated with its inhibitory action toward cell motility in NSCLC cells is not defined, and we attempted to identify this unique motif. First, *HOXA9* protein-coding sequences were subdivided into 3 regions with overlapping parts as shown in Figure 1A.

These coding sequences were amplified using PCR and cloned into an expression vector to test the functionality of partial *HOXA9* protein. The cloned N-terminal fragment included sequences from amino acids 1 to 104. The middle fragment consisted of 102 amino acid sequences (92 to 194 AA), and the C-terminal fragment included sequences from amino acid 182 to 272 of *HOXA9* protein. To synthesize proteins derived from these designated partial fragment sequences of *HOXA9*, start and stop codons were added in primer sets (primers are listed in Appendix A). Expression of the cloned *HOXA9* fragments in A549 and NCI-H1299 cells was confirmed by RT-PCR method using primers indicated in Appendix A (Figure 1B). We then performed in vitro invasion assay with Boyden transwell chambers in A549 and NCI-H1299 cells transfected with these expression vectors to determine the motif inhibiting the invasion of NSCLC cells. The images shown in the figure panels were representative of three independent experiments (Figure 1C–H). C-terminal fragment, *HOXA9*-C, exhibits more inhibition of cell invasion compared to control and other sequence motif proteins (*HOXA9*-N, -M) (Figure 1I) in A549 and NCI-H1299 cells. This result suggests that the 182–272 amino acid sequence, including *HOXA9* homeobox domain (HD; 209–262 AA), plays an important role in the inhibition of cell invasion in NSCLC cells.

### 3.2. The HOXA9-C Fragment Regulates Expression of EMT-Related Genes in NSCLC Cells

To explore the molecular mechanism driving changes in cell invasion by overexpression of the *HOXA9* fragments, we investigated the effect of *HOXA9* partial fragments on the regulation of *CDH1* or *SNAI2* expression in NSCLC cells. The expression level of *CDH1* was increased by overexpression of *HOXA9* full-length (*HOXA9*-F) and C-terminal fragment (*HOXA9*-C; 182–272 aa) in A549 and NCI-H1299 cells (Figure 2A).

Interestingly, the *HOXA9*-C fragment highly elevated *CDH1* gene level, as represented by the results of the invasion assay. On the contrary, *SNAI2* expression was reduced in *HOXA9*-F and *HOXA9*-C transfected A549 and NCI-H1299 cells (Figure 2B). The inverse correlation between *CDH1* and *SNAI2* expression induced by ectopic expression of the *HOXA9*-C fragment containing homeobox domain was observed in both A549 and NCI-H1299 cells. We also confirmed the expression of E-cadherin and SLUG proteins in NCI-H1299 cells (Appendix A). Based on these results, we investigated whether ectopic expression of *HOXA9*-C fragment represses NF-kB activity in NSCLC cells, as transcription factor NF-kB acts upstream of *SNAI2* during EMT. Both full-length *HOXA9* and *HOXA9*-C fragment suppressed NF-kB responsive reporter activity in A549 and NCI-H1299 cells (Figure 2C). We also examined the effect of *HOXA9*-C fragment on the direct binding of NF-kB to the promoter of *SNAI2* by performing ChIP assays in A549 cells. Our results showed that the ectopic expression of the *HOXA9*-C fragment suppressed the binding of NF-kB to the *SNAI2* promoter region (Figure 2D). These data suggest that the C-terminal fragment of *HOXA9* may play an important role in the *HOXA9*-mediated inhibitory process of cell invasion in NSCLC cells.

### 3.3. Intracellular Transduction of Synthetic Cell-Penetrating Peptide HADP (HOXA9 Active Domain Peptide)

LaRonde-LeBlanc [9] previously described a model of the *HOXA9*-DNA complex as illustrated in Figure 3A. *HOXA9* protein contains a homeobox domain from amino acid 209 to 262, which is a motif that directly drives the transcription of target genes. In our study, we found that the *HOXA9*-C fragment, containing the homeobox domain, can play an important role in EMT-related gene expression and cell invasion processes. To further validate these results, we synthesized a *HOXA9* active domain peptide (HADP) by including three sequence motifs present in the *HOXA9*-C fragment. The first motif is the linker motif (202–206 AA) of *HOXA9* protein, the second is the N-terminal arm motif (207–212 AA) contacted in DNA minor groove, and the third is the α-helix motif (246–260 AA) present in the DNA major groove.

As shown in Figure 3B, we visualized the docking structure between the synthesized peptide HADP and DNA using PyMOL molecular visualization software [24]. However, the peptides are unable to permeate the plasma membrane into the cell. So, we linked cell-permeable peptides (CPPs) such as polyarginine (R10) or CPP33 to the N-terminal region of the HADP and labeled it with a fluorescein isothiocyanate (FITC) to determine the cell membrane permeability. Both R10-HADP and CPP33-HADP were observed in the intracellular cytoplasm, however, the control C-HADP with no linked CPPs was not detected in the cytoplasm (Figure 3C). We also performed FACS analysis for quantifying intracellular cell-permeable peptides in A549 and NCI-H1299 cells. The synthetic peptide CPP33-HADP exhibited increased permeability in both cell lines compared to C-HADP (Figure 3D). Next, to investigate cell viability and cytotoxicity due to these peptides, we performed an MTT assay in a normal lung fibroblast cell line (MRC-5) and two non-small cell lung cancer cell lines (A549, NCI-H1299). As shown in Figure 3E, both R10-HADP and CPP33-HADP did not affect cellular viability.

### 3.4. CPP33-HADP Inhibits Invasion Ability of NSCLC Cells in Both In Vitro and In Vivo

To investigate whether the synthetic cell-penetrating peptide CPP33-HADP inhibits the invasion ability of NSCLC cells, we first performed an in vitro invasion assay. As shown in Figure 4A,B, synthetic peptide CPP33-HADP reduced the invasion ability of A549 and NCI-H1299 cells.

Hence, we developed an in vivo mouse model of lung metastasis to identify the role of this peptide in metastasis. A549 cells stably expressing luciferase were injected into the tail vein of mice and then administered an intraperitoneal injection of either CPP33-HADP or control peptide. A substantial reduction of tumor lesion-related symptoms was observed in animals treated with CPP33-HADP, compared to animals treated with control peptide (Figure 4C). To verify this data, we performed hematoxylin and eosin (H&E) staining of lung tissues in metastatic mice. Lung tissues of mice treated with control peptides showed visible tumor nodules. However, CPP33-HADP treatment resulted in decreased tumor nodules in lung tissues (Figure 4D). We also found that lung tissues of mice treated with CPP33-HADP exhibited reduced expression of SLUG while HADP was induced (Figure 4D and Appendix A). Taken together, we conclude that synthetic cell-permeable peptide CPP33-HADP may serve as a valuable pharmacological tool to prevent NSCLC metastasis.

## 4. Discussion

*HOXA9* is reported to play distinct roles in oncogenesis in acute myeloid leukemia, glioblastoma, and ovarian cancer and tumor suppression in breast, cervical, and hepatocellular cancer [25,26,27,28,29,30]. In NSCLCs, however, *HOXA9* has been implicated in the inhibition of tumor progression and metastasis [7,10,31]. In a previous study, we transduced recombinant cell-permeable *HOXA9* protein conjugated with CPP directly into cells to confirm their function in NSCLCs. We demonstrated that recombinant cell-permeable *HOXA9* protein inhibits cell motility of NSCLCs in vitro and in vivo [22]. However, the key domain of the *HOXA9* protein responsible for inhibiting cell motility is currently undefined. Therefore, to develop a system to deliver the key inhibitory domain of *HOXA9*, we cloned three partial DNA fragments of *HOXA9* and analyzed its role in the invasion process of tumor cells. Amongst the fragments, upregulation *HOXA9*-C fragment (182–272 AA) inhibited invasion ability of NSCLCs similar to full-length *HOXA9*. EMT is a key event in the initiation of metastasis in tumor cells. Induction of the transcription factor such as SNAIL, SLUG, and TWIST 1, and/or suppression of cell-cell adhesion protein E-cadherin are strongly implicated in the process of EMT induction [11]. In our study, ectopic overexpression of *HOXA9*-C fragment resulted in upregulated expression of *CDH1* but downregulated expression of *SNAI2* in A549 and NCI-H1299 cells (Figure 2). According to previous reports, transcription factor NF-kB plays an important role in EMT and metastasis in NSCLC [32]. In our study, the transcriptional activity of NF-kB was reduced by the overexpression of the *HOXA9*-C fragment. This result indicates that the C-terminal region of *HOXA9* may be a key motif for the prevention of EMT-mediated cell invasion.

*HOX* genes are known to be differentially expressed in various solid tumors compared to their respective normal tissues [6]. These differences in gene expression appear to be related to tumorigenesis. Members of the HOX family act as oncogenes or tumor suppressors in numerous tissues. The multiple functions of HOX can be attributed to slight differences in the homeobox domain of *HOX* genes [5]. In addition, several studies showed that *HOX* genes influence the NF-ĸB pathway by modulating the expression or the stability of upstream regulators. Oncogenic HOX proteins (HOXA1, HOXB7, HOXB13, and HOXC10) have been reported to stimulate the NF-ĸB pathway. However, *HOXA9* has also been shown to suppress the NF-ĸB pathway by inhibiting the DNA binding ability of NF-ĸB [33]. *HOXA9* is well-known as a transcription factor containing a DNA binding domain (HD, homeobox domain) from amino acids 209 to 262. LaRonde-LeBlanc and Wolberger have reported the structure of *HOXA9* complexed with DNA, showing hydrogen bonds between K58 of *HOXA9*-HD and S43 of PBX1 protein [9]. Slattery et al. demonstrated that HOX/transcription activator-like effector (TALE) complex increased DNA-binding specificity and affinity compared to HOX monomer binding [34]. Recently, the interaction between HD of *HOXA9* protein and TALE-class cofactors, PBX1 and MEIS, has been reported. The study demonstrated that a specific amino acid of the HD region of *HOXA9* protein and hexapeptide (HX) motif of *HOXA9* protein plays an important role in forming a complex with TALE-class cofactors [35]. In this study, we synthesized 36-mer HADP including three sequence motifs present in the *HOXA9*-C fragment. However, proteins or peptides cannot generally penetrate across cell the membrane due to their size and polarity. To combat this obstacle, we chose to use CPPs as a delivery agent. CPPs are broadly classified into non-tissue specific and tissue-specific groups. Non-tissue-specific CPPs are subdivided into three types including cationic peptides, hydrophobic peptides, and amphipathic peptides [14]. CPP33 is a specific CPP for lung cancer-specific delivery [16]. In this study, we synthesized cell-penetrating peptides fused with CPP33 for delivery to lung tissues as well as a non-tissue specific 10-mer homo-polymer of arginine (R10). The penetration of these peptides in NSCLC cells was confirmed by using confocal and FACS analysis methods. The transduction efficiency of CPP33-HADP was preferable to that of R10-HADP in A549 and NCI-H1299 cells.

In analyzing the role of a synthetic cell-permeable peptide in the motility process, our data indicate that CPP33-HADP can inhibit the invasion ability of NSCLC cells. Therefore, we suggest that the synthetic peptide CPP33-HADP may have essential application as a therapeutic agent to prevent metastatic progression of NSCLC. The efficacy of CPP-mediated antitumor strategies is demonstrated in tumor cell lines and animal xenograft models [36], and they are in phase I clinical trials for patients with advanced solid tumors [37]. However, CPP-conjugated drugs are still not FDA approved for reasons such as in vivo stability, immunogenicity issues, poor efficiency, and toxicity [38]. Indeed, further studies on CPP33-HADP, focusing on stability, immunogenicity, efficiency, and toxicity of the synthetic peptide, are warranted before its application as a therapeutic drug.

In conclusion, we found a key motif of *HOXA9* protein, which prevents cell invasion in A549 and NCI-H1299 cells. The CPP-conjugated HADP markedly inhibits the invasion ability of NSCLC cells in in vitro and in vivo experiments. These findings strongly suggest that the synthetic peptide CPP33-HADP has a significant potential for therapeutic applications in metastatic lung cancer.

## Figures and Tables

**Figure 1 biomolecules-10-01589-f001:**
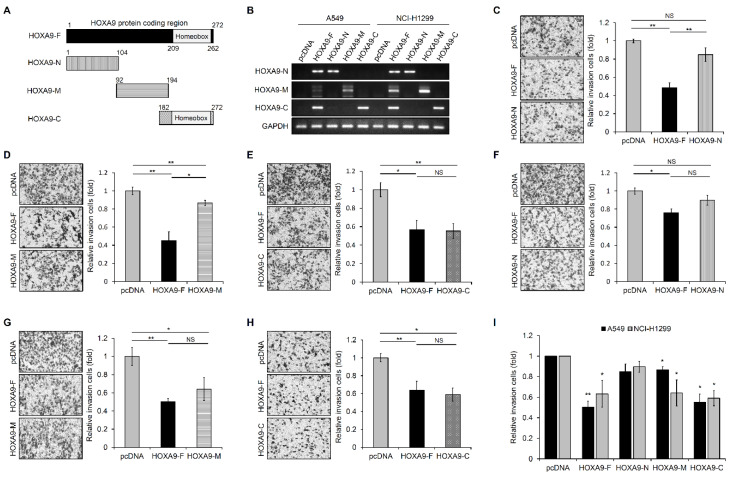
C-terminal fragment of Homeobox A9 (*HOXA9*) inhibits invasion of non-small cell lung cancer (NSCLC) cells. (**A**) Schematic representation of the cloned *HOXA9* partial fragments. (**B**) The expression level of indicated partial fragments cloned into the expression vector. The expression was confirmed by the RT-PCR method in transfected A549 or NCI-H1299 cells. (**C**–**E**) Invasion assay with Boyden transwell chambers in the *HOXA9* partial fragments transfected A549 cells. (**F**–**H**) Invasion assay in the *HOXA9* partial fragments transfected NCI-H1299 cells. All images are shown at 200× magnification. (**I**) Relative invasion values were obtained from all figure panels (**C**–**H**). All data represent the mean ± SD from three independent experiments. * *p* < 0.05; ** *p* < 0.01.

**Figure 2 biomolecules-10-01589-f002:**
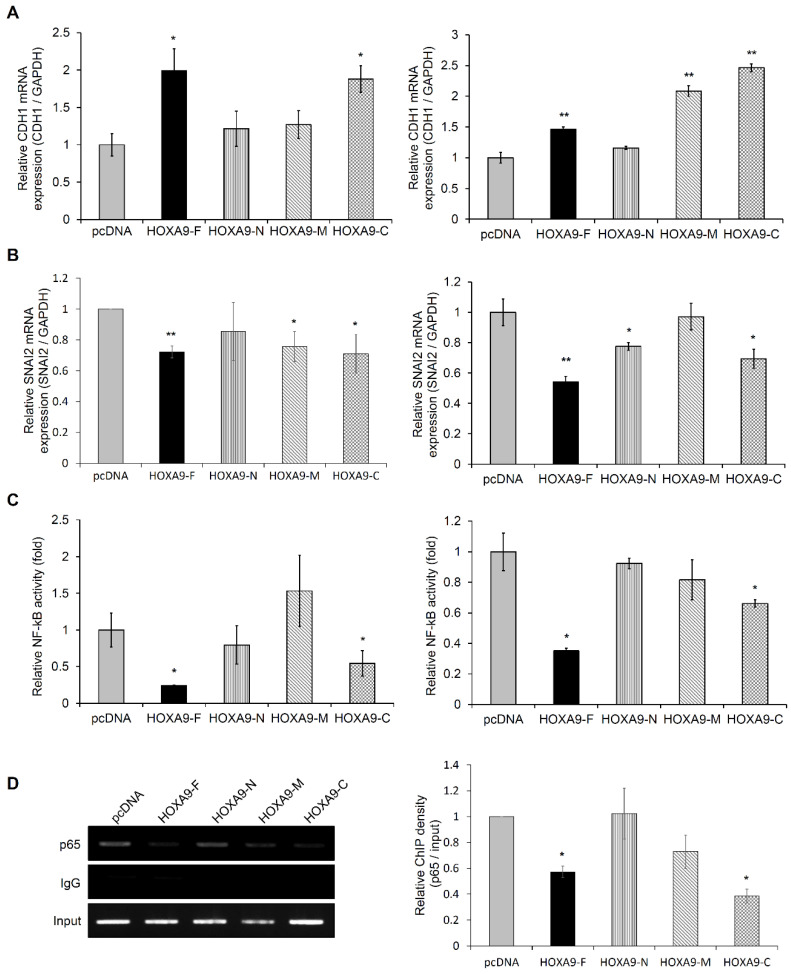
C-terminal fragment of *HOXA9* regulates the expression of Epithelial-to-Mesenchymal Transition (EMT)-related genes. (**A**) The effect of *HOXA9* full-length or partial fragments on change of *CDH1* mRNA expression in A549 (left) or NCI-H1299 (right) cells. The expression of *CDH1* was analyzed by qRT-PCR. (**B**) The effect of *HOXA9* full-length or partial fragments on *SNAI2* mRNA expression in A549 (left) or NCI-H1299 (right) cells. (**C**) The NF-kB activity in *HOXA9* full-length or partial fragments transfected A549 (left) or NCI-H1299 (right) cells. The activity was measured by luciferase-based reporter assays. (**D**) Identification of NF-kB binding on the promoter region of *SNAI2* by Chromatin Immunoprecipitation (ChIP) method in *HOXA9* full-length or partial fragments transfected A549 cells. All data represent the mean ± SD from three independent experiments. * *p* < 0.05; ** *p* < 0.01.

**Figure 3 biomolecules-10-01589-f003:**
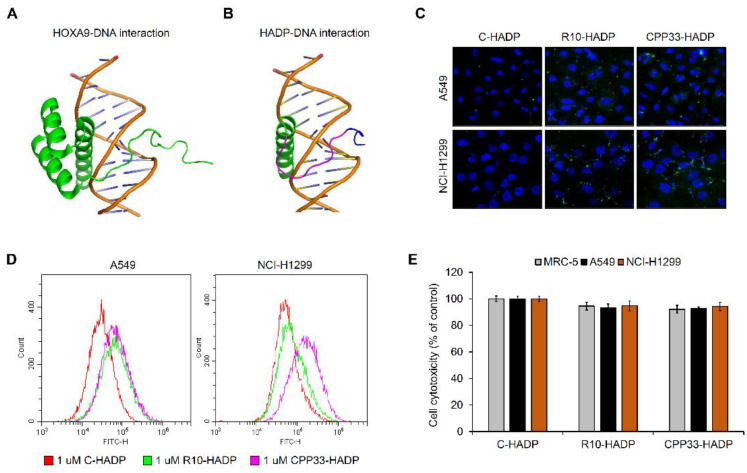
Structure and intracellular transduction of synthetic cell-permeable peptide HADP. (**A**) The *HOXA9*-DNA complex was reported by LaRonde-LeBlanc (2003). (**B**) Docking simulation between *HOXA9*-HADP and DNA using PyMOL analysis. Blue indicates the linker motif of *HOXA9* protein, and red represents the N-terminal arm motif of *HOXA9* protein. Green refers to the a-helix motif of *HOXA9* protein. (**C**) Visualization of intracellular cell-penetrating peptides using immunofluorescence microscopy. All images are shown at 200× magnification. (**D**) Estimation of intracellular cell-permeable peptides using FACS analysis. (**E**) Determination of cell cytotoxicity using an MTT assay in cell-permeable peptides treated cells.

**Figure 4 biomolecules-10-01589-f004:**
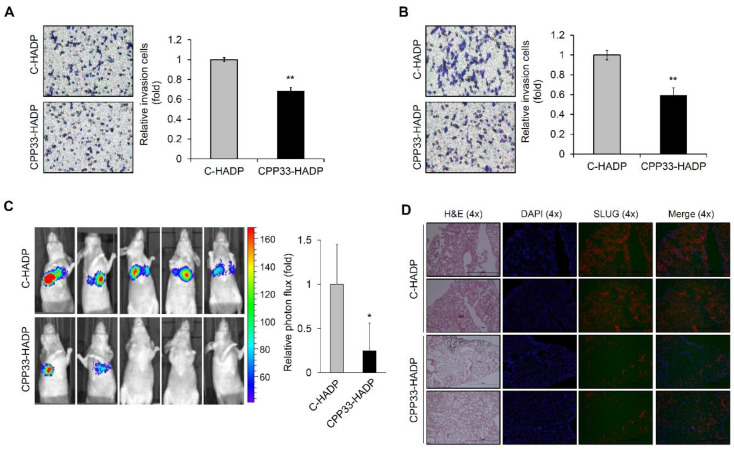
Synthetic peptide CPP33-HADP inhibits the invasion ability of NSCLC cells. (**A**,**B**) Cell invasion assay using a Boyden chamber assay in C-HADP and CPP33-HADP treated A549 and NCI-H1299 cells. All data represent the mean ± SD from three independent experiments. ** *p* < 0.01. (**C**) The effect of C-HADP or CPP33-HADP on in vivo cell motility assay using luciferase-expressed A549 cells. Tumor volume measured by bioluminescence in lung tissue of nude mice. * *p* < 0.05. (**D**) H&E staining and SLUG expression in C-HADP or CPP33-HADP treated xenograft lung tissues.

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
