# Peer review of "A Synthetic CPP33-Conjugated HOXA9 Active Domain Peptide Inhibits Invasion Ability of Non-Small Lung Cancer Cells"

_biomolecules, 2020, doi:10.3390/biom10111589_

Round 1

Reviewer 1 Report

I have found the findings original and interesting, supported by the experimental evidence.

Author Response

I have found the findings original and interesting, supported by the experimental evidence.

(Answer) We thank the reviewer for his/her thorough and insightful review of our manuscript.

Reviewer 2 Report

This is an extension of authors’ previous report on HOXA9 upregulation inhibited NF-kappaB-mediated SNAI2 and CDH1 expression. Authors tried to pinpoint the region responsible for the activity of HOXA9. It is very interesting topic and may be very useful for future drug development. However, I found authors did not provide enough background information for us to completely understand the whole manuscript. In addition, there are some inconsistent in term usage. I just list some of the questions.

  1. The genomic DNA source for PCR amplification of the three fragments of HOXA9 should be specifically mentioned.
  2. The primers for RT-PCR verification of expression shown in Fig. 1B are missing?
  3. Figure 1A homeobox domain region (aa206-265) is different from line 197: aa209-262. Please be consistent.
  4. Homeobox is the DNA binding domain which is essential for the transcription factor activity of HOXA9. From Figure 1, authors found that C-term 182-272 exerted the same anti-inverse effect as full length HOXA9. Would it be possible that removing N-terminal end of HOXA9 would exert adverse effect which was not detected in this assay?
  5. Authors only provided mRNA expression data of CDH1, but protein data is preferred.
  6. The same as CDH1 applied to SNAI2. SNAI2 is a transcription factor and its expression and activity should be measured in nuclear protein as well.
  7. From the literature, HOXA9 is a transcription factor for E-selectin and VCAM-1 expression. Is it possible that HOXA9 also activates these two gene expression in A549 or NCI-H1299 cells?
  8. Figure 2C: the activity analysis of NK-kappB stated in section 2.5 differed from what present in Figure 2C. Please correct the description.
  9. Figure 2D: ChIP experiment was conducted in A549 or NCI-H1299 cells, please specify.
  10. It has been reported that NF-κB activity is required for the expression of Twist1/Twist, SNAI2/Slug, and ZEB2/Sip1 to drive EMT and CIC phenotypes in NSCLC. I just wondered whether HOXA9-mediated NF-kappaB down-regulation also caused the repression of other 2 genes?
  11. It is unclear how and why these three peptides, C-HOXA9-HDAP, R10-HOXA9-HDAP and CPP33-HOXA9-HADP, were designed. Authors should give more background information regarding their structures and characteristics in the introduction before getting into materials and method or results.
  12. 3: line 244-247, there were three different domain peptides synthesized by authors. However, we did not find the relevant results in Figure 3? This section should be rewrite to clarify the results.
  13. It seems that only homeobox domain is required for the anti- NSCLC activity? If so, authors may want to compare the differences as compared with other HOX? And discuss why only HOXA9 ,but not other HOXs, has this specific activity?
  14. Line 97-100 and section 3.3 and 3.4 used different terms for the peptides, please be consistent.

Author Response

Responses to the Reviewer’s comments:

Reviewer: 2

This is an extension of authors’ previous report on HOXA9 upregulation inhibited NF-kappaB-mediated SNAI2 and CDH1 expression. Authors tried to pinpoint the region responsible for the activity of HOXA9. It is very interesting topic and may be very useful for future drug development. However, I found authors did not provide enough background information for us to completely understand the whole manuscript. In addition, there are some inconsistent in term usage. I just list some of the questions.

  1. The genomic DNA source for PCR amplification of the three fragments of HOXA9 should be specifically mentioned.

(Answer) We appreciate the reviewer's suggestion. In accordance with this suggestion, we have included the sentences “HOXA9 gene contains two exons and one intron. The HOXA9-N and M primer set is located at the exon 1, and the HOXA9-C primer set corresponds to the interface region between the two exons including the intron” in the Materials and Methods section. The locations of primers are illustrated below:

We have attached word file including figure.

  1. The primers for RT-PCR verification of expression shown in Fig. 1B are missing?

(Answer) We are thankful for the reviewer's observation. Accordingly, we have inserted the sentence “using primers indicated in supplementary table 1” in the Results section.

  1. Figure 1A homeobox domain region (aa206-265) is different from line 197: aa209-262. Please be consistent.

(Answer) We apologize for the error. We have changed the homeobox domain region from “206-265” to “209-262” in figure 1A.

  1. Homeobox is the DNA binding domain which is essential for the transcription factor activity of HOXA9. From Figure 1, authors found that C-term 182-272 exerted the same anti-inverse effect as full length HOXA9. Would it be possible that removing N-terminal end of HOXA9 would exert adverse effect which was not detected in this assay?

(Answer) We thank the reviewer for the insightful comment. Invasion assays in cells overexpressing partial fragments of HOXA9 indicated that HOXA9-C, containing the homeobox domain, exhibited more inhibition compared to the other fragments. However, HOXA9-M, in which the N- and C-terminus of HOXA9 are deleted, has a similar effect to that of the control group. Overexpression of the N-terminal fragment of HOXA9 also did not inhibit cell invasion compared to the control. Although we could not perform this experiment in time, these results demonstrate that removing the N-terminal end of HOXA9 does not affect the inhibition of cellular invasion in NSCLC cells.

  1. Authors only provided mRNA expression data of CDH1, but protein data is preferred

(Answer) We sincerely appreciate the reviewer’s suggestion. We agree with the reviewer’s assessment, and therefore performed western blot analysis to show the expression pattern of E-cadherin protein in NCI-H1299 cells treated with HOXA9 partial fragments. We have included the protein expression data in the supplementary figure 1.

We have attached word file including figure.

  1. The same as CDH1 applied to SNAI2. SNAI2 is a transcription factor and its expression and activity should be measured in nuclear protein as well.

(Answer) We sincerely appreciate the reviewer’s suggestion. We performed western blot analysis to show the expression pattern of SLUG protein in NCI-H1299 cells treated with HOXA9 partial fragments. However, as we wanted to identify the change in SNAL2 expression by the HOXA9-NF-ĸB axis, we were unable to confirm SNAL2 expression and activity in the nucleus. Thus, we have included the total SLUG protein expression data in the supplementary figure 1.

We have attached word file including figure.

  1. From the literature, HOXA9 is a transcription factor for E-selectin and VCAM-1 expression. Is it possible that HOXA9 also activates these two gene expression in A549 or NCI-H1299 cells?

(Answer) We are grateful for the reviewer’s suggestion. Accordingly, we performed RT-PCR analysis to show the expression pattern of E-selectin and VCAM-1 in both A549 and NCI-H1299 cells treated with HOXA9 partial fragments. However, no significant change in expression of E-selectin and VCAM-1 was observed in the presence of HOXA9 partial fragments. The expression data is illustrated below:

We have attached word file including figure.

  1. Figure 2C: the activity analysis of NK-kappB stated in section 2.5 differed from what present in Figure 2C. Please correct the description

(Answer) We thank the reviewer for carefully reviewing our manuscript. We have replaced the original sentence “Cells were harvested for luciferase assays after incubation with 1 µM C-HOXA9-HAD, R10-HOXA9-HAD, or CPP33-HOXA9-HAD peptide for 24 h” with “Cells were harvested for luciferase assays after transfection with pcDNA, HOXA9-F, HOXA9-N, HOXA9-M, or HOXA9-C for 24 h.” on line 131-132 of the Materials and Methods section.

  1. Figure 2D: ChIP experiment was conducted in A549 or NCI-H1299 cells, please specify

(Answer) We have now included the name of the cell line “A549” in the legend of Figure 2D.  

  1. It has been reported that NF-κB activity is required for the expression of Twist1/Twist, SNAI2/Slug, and ZEB2/Sip1 to drive EMT and CIC phenotypes in NSCLC. I just wondered whether HOXA9-mediated NF-kappaB down-regulation also caused the repression of other 2 genes?

(Answer) We thank the reviewer for this comment. As per your suggestion, we performed RT-PCR analysis to determine the expression pattern of Twist1 and ZEB2 in A549 and NCI-H1299 cells treated with HOXA9 partial fragments. The difference in expression of Twist1 and ZEB2 upon HOXA-9 full-length or partial fragments overexpression is relatively low compared to that of SNAI2. The expression data is illustrated below:

We have attached word file including figure.

  1. It is unclear how and why these three peptides, C-HOXA9-HDAP, R10-HOXA9-HDAP and CPP33-HOXA9-HADP, were designed. Authors should give more background information regarding their structures and characteristics in the introduction before getting into materials and method or results

(Answer) We deeply appreciate the reviewer’s suggestion for improving our manuscript. In accordance with this suggestion, we have inserted the sentences “Previous studies have found that small differences in homeodomains themselves confer unique properties to HOX genes [5]; It is well established that HOXA9 contains a DNA-binding homeobox domain and regulates downstream gene expression [9]. Polyarginine CPP uptake is reported in major tissues including, bladder, kidney, liver, and lung, using intraperitoneal or intravenous injections [15].” respectively, in the Introduction section.

  1. line 244-247, there were three different domain peptides synthesized by authors. However, we did not find the relevant results in Figure 3? This section should be rewrite to clarify the results.

(Answer) We thank the reviewer for this pertinent suggestion. In accordance with this suggestion, we marked the locations of the three different domains on the synthesized peptide and incorporated the sentences “Blue indicates the linker motif of HOXA9 protein, and red represents the N-terminal arm motif of HOXA9 protein. Green refers to the a-helix motif of HOXA9 protein” into figure 3B.

We have attached word file including figure.

  1. It seems that only homeobox domain is required for the anti- NSCLC activity? If so, authors may want to compare the differences as compared with other HOX? And discuss why only HOXA9 ,but not other HOXs, has this specific activity?

(Answer) We deeply appreciate the reviewer’s comment. In accordance with this suggestion, we have inserted the sentences “HOX genes are known to be differentially expressed in various solid tumors compared to their respective normal tissues [6]. These differences in gene expression appear to be related to tumorigenesis. Actually, members of the HOX family act as oncogenes or tumor suppressors in numerous tissues. The multiple functions of HOX can be attributed to slight differences in the homeobox domain of HOX genes [5]. In addition, several studies showed that HOX genes influence the NF-ĸB pathway by modulating expression or the stability of upstream regulators. Oncogenic HOX proteins (HOXA1, HOXB7, HOXB13, and HOXC10) have been reported to stimulate the NF-ĸB pathway. However, HOXA9 has also been shown to suppress the NF-ĸB pathway by inhibiting DNA binding ability of NF-ĸB [33]. ” in the Discussion section.

  1. Line 97-100 and section 3.3 and 3.4 used different terms for the peptides, please be consistent.

(Answer) We thank the reviewer for carefully reviewing our manuscript. We have modified the terms of peptide from “C-HOXA9-HADP, R10-HOXA9-HADP, and CPP33-HOXA9-HADP” to “C-HADP, R10-HADP, and CPP33-HADP,” respectively, in the Materials and Methods section.

Reviewer 3 Report

The manuscript by Seong-Lan Yu and his coauthors is based on previous HOXA9 reports in lung cancer and tried to identify the domain on HOXA9 exerting the tumor-suppressive function. In this report, the authors found the 182-272 amino acid sequence containing homeobox domain is crutial for the tumor-suppressive role of HOXA9. Also, the authors link CPP33 peptide to HOXA9-C fragment for enhancing the penetration of the cell membrane and validated the role HOXA9 in vivo.
Overall, this is an interesting study for the improvement of HOXA9 in clinical application. However, this study lack enough details to prove their conclusion. There is no western blot experiment in this study. This is the most lethal weakness of this study. The comments below may be helpful to improve this study.
1. In Fig. 1, there is no western blot experiments to show the successful expression of HOXA9 fragments. HOXA9 functions in its protein form. The detection of HOXA9 RNA expression by qPCR is not enough. If the authors want to prove their conclusion, they must show the expression of HOXA9 full-length and fragments using western blot.
2. In Fig. 2, the same problem should be solved by showing the western blot results of CDH11, SNAI2, HOXA9 full-length and fragments proteins.
3. In Fig. 4, the expression of C-HADP and CPP33-HADP peptides should be detected in tumor samples using both western blot and IHC.

Author Response

Responses to the Reviewer’s comments:

Reviewer: 3

The manuscript by Seong-Lan Yu and his coauthors is based on previous HOXA9 reports in lung cancer and tried to identify the domain on HOXA9 exerting the tumor-suppressive function. In this report, the authors found the 182-272 amino acid sequence containing homeobox domain is crutial for the tumor-suppressive role of HOXA9. Also, the authors link CPP33 peptide to HOXA9-C fragment for enhancing the penetration of the cell membrane and validated the role HOXA9 in vivo.

Overall, this is an interesting study for the improvement of HOXA9 in clinical application. However, this study lack enough details to prove their conclusion. There is no western blot experiment in this study. This is the most lethal weakness of this study. The comments below may be helpful to improve this study.

  1. In Fig. 1, there is no western blot experiments to show the successful expression of HOXA9 fragments. HOXA9 functions in its protein form. The detection of HOXA9 RNA expression by qPCR is not enough. If the authors want to prove their conclusion, they must show the expression of HOXA9 full-length and fragments using western blot.

(Answer) We thank the reviewer for the insightful comments. We did not tag the partial fragment sequences of HOXA9 in the expression vector to rule out the possibility of changes in protein function due to the tag sequences. Currently, HOXA9 antibody (07-178, EMD Millipore) for western blotting detects the 194-272 aa site, so it can only identify HOXA9-F and HOXA9-C, but not HOXA9-N and HOXA9-M. In addition, we faced difficulties in detecting HOXA9-C in western blotting because it is less than 10 kDa in size. Thus, we confirmed the expression of full-length HOXA9 and HOXA9 fragments using RT-PCR. Naturally, the efficacy of the cloned expression vector was confirmed by detecting the expression of full-length HOXA9 in the transfected cells.

We have attached word file including figure

  1. In Fig. 2, the same problem should be solved by showing the western blot results of CDH11, SNAI2, HOXA9 full-length and fragments proteins.

(Answer) We appreciate the reviewer’s pertinent suggestion for improving our manuscript. As per your suggestion, we performed western blot analysis to determine the expression pattern of E-cadherin and SLUG proteins in NCI-H1299 cells treated with HOXA9 partial fragments and have included the expression data in supplementary figure 1.

We have attached word file including figure

  1. In Fig. 4, the expression of C-HADP and CPP33-HADP peptides should be detected in tumor samples using both western blot and IHC.

(Answer) We appreciate the reviewer's suggestion. It is challenging to efficiently detect small peptides by the conventional western blotting method because the peptides readily detach from the blotted membranes. Thus, we used fluorescence microscope to determine the expression of peptides in tumor samples treated with C-HADP or CPP33-HADP. In the experiment, different deparaffinized tissues were used because the manuscript's H&E stained tissues generated autofluorescence with a high background. So we have included the microscope images in the supplementary figure 2.

We have attached word file including figure

Round 2

Reviewer 3 Report

The authors have addressed my concerns and the quality of this manuscript has been improved.